# Characterizing Continual Learning Scenarios for Tumor Classification in Histopathology Images

**Veena Kaustaban**[1]                              veena.kaustaban@roche.com
**Qinle Ba**[1]                                         qinle.ba@roche.com
**Ipshita Bhattacharya**[1*]           ipshita.sb.bhattacharya@gmail.com
**Nahil Sobh**[1]                       nahil.sobh@contractors.roche.com
**Satarupa Mukherjee**[1]               satarupa.mukherjee@roche.com
**Jim Martin**[1]                         jim.martin@contractors.roche.com
**Mohammad Saleh Miri**[1]                     saleh.miri@roche.com
**Christoph Guetter**[1]                   christoph.guetter@roche.com
**Amal Chaturvedi**[1]                     amal.chaturvedi@roche.com
[1]*Roche Sequencing Solutions, Santa Clara, CA*

**Editors:** Under Review for MIDL 2023

## Abstract

Deep-learning models have achieved unprecedented performance in fundamental computational tasks in digital pathology (DP) based analysis, such as image classification, cell detection and tissue segmentation. However, such models are known to suffer from catastrophic forgetting when adapted to unseen data distribution with transfer learning. With an increasing need for deep-learning models to handle ever-changing data distributions, including evolving patient population and new diagnosis assays, it is crucial to introduce methods for alleviating the such model forgetting. To this end, continual learning (CL) models are promising candidates. However, to our best knowledge, there's no systematic study of CL models in DP-specific applications. Here, we propose various CL scenarios in DP settings, where histopathology image data from different sources/distributions arrive sequentially, the knowledge of which is integrated into a single model without training all the data from scratch. To benchmark the performance of recently proposed continual learning algorithms in the proposed CL scenarios, We augmented a dataset for colorectal cancer H&E classification to simulate shifts of image appearance and evaluated CL methods on this dataset. Furthermore, we leveraged a breast cancer H&E dataset along with the colorectal cancer dataset to assess continual learning methods for learning from multiple tumor types. We revealed promising results of CL in DP applications, potentially paving the way for application of these methods in clinical practice.

**Keywords:** Digital Pathology, Continual Learning, Tumor Tissue Classification

## 1. Introduction

A general strategy to learn from multiple datasets considers training all the existing and newly arrived data from scratch, leading to increasingly high demand of computing resource, time and data storage. On the other hand, updating a model with transfer learning is known to be ineffective due to catastrophic forgetting (Kirkpatrick et al., 2017). On the contrary, continual learning algorithms (Kirkpatrick et al., 2017; Chaudhry et al., 2018; Lopez-Paz and Ranzato, 2017; Rebuffi et al., 2017) aim at efficiently and effectively adapting a model

---

* Work done at Roche

to new data streams without forgetting of learned knowledge. These algorithms have largely been tested on less complex datasets from non-biomedical domains. Here, we report the first study to assess the feasibility of CL in DP-based applications (Kaustaban et al., 2022).

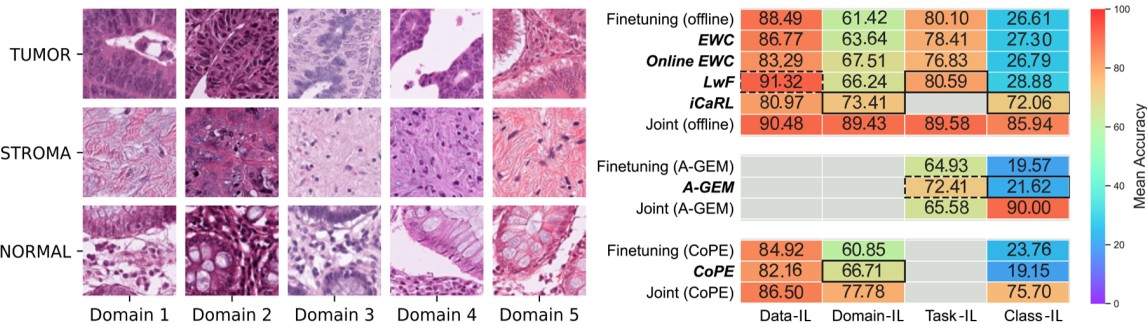

Figure 1: Left: Example augmented CRC images from 5 domains. Right: Test accuracy of offline and online methods on augmented CRC against their corresponding baselines.

## 2. Methods

**Dataset 1 - CRC.** A colorectal cancer (CRC) dataset (Kather et al., 2019) was used to generate simulated data streams, which comprise whole-slide H&E images (136 patients) with tile-wise class labels for 9 tissue types. We randomly selected 8700 images (7000 train; 2700 validation) from each class for training and 7150 images for testing. **Dataset 2 - PatchCam.** To characterize the effectiveness of CL methods on learning from multiple tissue types, PatchCam from (Bejnordi et al., 2017) was used for breast cancer classification after stain-normalization (Macenko et al., 2009), which has class labels of "normal" and "tumor" from 400 breast cancer H&E whole-slide images. **Augmented CRC - Simulating**

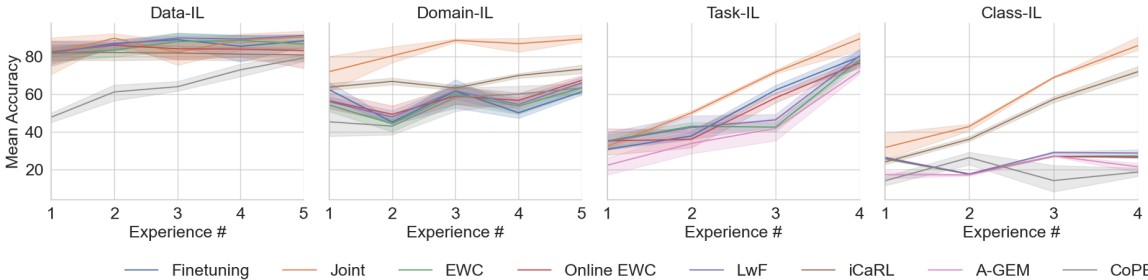

Figure 2: Online and offline CL performance at each experience against offline baselines.

**Domain Shifted Data Streams.** We generated an augmented CRC dataset to simulate the commonly observed variations of stain appearance from multiple data sources (Fig. 1 Left). Specifically, we first performed stain unmixing of CRC images and then applied augmentation in the optical density space before remixing the stain-unmixed intensity images. In addition to the original images (Domain 1), we generated images simulating different

dye concentrations for H&E stain (Domain 2: increased stain intensity), faded stains due to slide aging (Domain 3: decreased eosin intensity) and changes of reagent manufacturer, scanners or stainers (Domain 4: change in hue; Domain 5: change in hue and saturation). **Continual Learning with Augmented CRC.**: We identified four CL scenarios suitable for DP applications based on how new streams of data differ from the previous ones. Specifically, data streams contain (1) an equal number of images from all domains (Data-IL), (2) an equal number of images from one domain (Domain-IL), (3) images from new classes from all domains (Class-IL), or (4) images from new classes from all domains for each of the multi-task heads (Task-IL). Please see (Kaustaban et al., 2022) for more details. **Continual Learning Algorithms.**: Recently proposed CL methods can be largely classified into three categories: replay, regularization and parameter isolation (Vokinger et al., 2021).

- *Replay methods* select original images (Rebuffi et al., 2017), deep representations (Van de Ven and Tolias, 2019) or model-generated pseudo samples (Shin et al., 2017) via various heuristics, which are stored in memory and replayed at later learning stages to overcome forgetting. *iCaRL* stores a subset of most representative examples per class in memory. *A-GEM* constraints model updates by projecting the estimated gradient on the direction determined by randomly selected samples from a replay buffer. *CoPE* enables rapidly evolving prototypes with balanced memory and a loss function that updates the representations or pseudo prototypes.

- *Regularization methods* include a regularization term in the loss function to penalize model updates that could lead to large deviation from an existing model to avoid forgetting of learned knowledge. *EWC/Online EWC* includes a regularization term in the loss function to penalize large changes to network weights that are important for previous tasks. *LwF* distills knowledge from a previous model to an updated model trained on new data with an additional distillation loss term for replayed data.

- *Parameter isolation methods* assign different model parameters to each task head. Note that parameter isolation only applies to Task-IL.

## 3. Results

To our surprise, continual learning was most effective for the challenging Class-IL scenario (Figure 1 Right). Comparing offline CL methods, iCaRL had the best overall performance, even comparable to upper-bound joint training in the Class-IL scenario (Figure 1 Right: top block, Figure 2). For online CL (Figure 1 Right: lower blocks), which learned from hundreds of small data batches in a near continuous stream, we observed that A-GEM outperformed upper-bound baseline in Task-IL scenario (Figure 2). However, online methods did not outperform offline ones. We further assessed sequential learning from multiple tumor types in domain-IL for CRC and breast cancer datasets and found that (a) adding more examples and (b) "hard to easy curriculum" generated better results.

In summary, we found that continual learning for DP, despite being challenging, is feasible and promising. CL methods were computationally efficient, taking only about 28% of the runtime as joint training. Though patient data evolve quickly nowadays, FDA has not approved algorithms based on CL (Vokinger et al., 2021) and extensive research is needed to establish related regulations. Our evaluation approaches and proposed method to generate domain-shifted datasets can potentially serve as the first step towards this goal.

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
