# OpenReview forum: "Characterizing Continual Learning Scenarios for Tumor Classification in Histopathology Images"
_MIDL.io/2023/Short_Paper_Track — MIDL 2023 Short paper track Poster_

### Official Review · Reviewer_zewy · 2023-04-20
**Experimental-focused study with interesting insights**

**Rating:** 7
**Confidence:** 4

**Review:**

Summary
This work conducted an experiment-focused systematic study of applying continual learning (CL) methods to digital pathology (DP) specific applications. The experiments include investigating the performance of applying three categories of CL methods to four DP CL scenarios in a carefully designed DP data stream, additionally with comparisons with offline baselines.

Strengths

Rich carefully-designed experiments provide readers with good initial insights into how CL methods can be used in DP problems.

Weaknesses

The definition of the data-IL and domain-IL looks a little confusing, does domain-IL means there are clear boundaries for each domain in the data stream, and data-IL means all data of all domains are randomly mixing together in the stream? Some examples or figures (if there is space) could be very helpful.

---

### Official Review · Reviewer_bCa6 · 2023-04-20
**Review for paper 93**

**Rating:** 7
**Confidence:** 4

**Review:**

This is a very interesting which will interest the community , especially those working on digital pathology. Evaluating continual learning settings for digital pathology is a good contribution to the literature.